# Experimental Study with Plaster Mortars Made with Recycled Aggregate and Thermal Insulation Residues for Application in Building

Daniel Ferrández [1,*], Manuel Álvarez [1], Pablo Saiz [2] and Alicia Zaragoza [1]

1 Department of Building Technology, Polytechnic University of Madrid, Avenida Juan de Herrera, 6, 28040 Madrid, Spain; manuel.alvarezd@upm.es (M.Á.); alicia.zaragoza@alumnos.upm.es (A.Z.)
2 Department of Financial Economics, Accounting and Modern Language, Rey Juan Carlos University, Vicálvaro Campus, Paseo de los Artilleros, s/n, 28032 Madrid, Spain; pablo.saiz@urjc.es
* Correspondence: daniel.fvega@upm.es

**Abstract:** The high demand for natural resources and increased industrial activity is driving the construction sector to search for new, more environmentally friendly materials. This research aims to analyse plaster mortars with the incorporation of construction and demolition waste (CDW) to move towards a more sustainable building sector. Three types of aggregates (natural, recycled concrete and recycled from ceramic walls) and two types of insulation waste (expanded polystyrene with graphite and mineral wool) have been added to the plaster matrix to evaluate its mechanical and physical properties and its suitability in the elaboration of prefabricated materials. The results show how plaster mortars made with recycled aggregates have higher mechanical resistance than conventional plaster without incorporating sand. The incorporation of crushed mineral wool residues improves the flexural strength of plaster mortars and their application in the execution of prefabricated panels. Likewise, the expanded polystyrene residues reduce the final density of mortars, improving their behaviour against water absorption and reducing the final thermal conductivity of plaster material.

**Keywords:** plaster mortar; recycled aggregates; thermal insulation; building

## 1. Introduction

The construction sector is included in the six factors (cropland, grazing land, fishing grounds, forest products, carbon and built-up land) that make up the ecological footprint of humanity [1]. For this reason, it is necessary to involve efforts to develop new construction materials based on circular economy criteria [2]. To this end, the European Union has defined a firm line of action that is included in the "European Green Deal" presented by the Commission Communication of 11 December 2019 [3]. This document includes as a main element the efficient use of resources in construction, as well as ensuring that less waste is produced. For this reason, more and more researchers are channelling their studies towards the scope of sustainable construction and the use of raw materials from construction and demolition waste (CDW) management [4,5].

These CDW are the main source of waste generation in modern society [6]. Thus, the application of the 3Rs principle (reduce-reuse-recycle) avoids the generation of waste and converts it back into resources, thus closing the circle in industrial ecosystems [7]. With regard to the composition of these CDW, according to reports published by the European Commission [8], most of the waste comes from the demolition of concrete structures (12–40%) and factory works made with ceramic materials (8–54%), and the main recycling process of these residues is the generation of aggregates for construction [9]. Additionally, the volume of waste from insulating materials is growing due to European initiatives to improve the energy efficiency of buildings and their rehabilitation [10].

Concrete and ceramic material waste are generally inert and low polluters; however, they occupy a large volume in landfills and have a strong impact on waste management in cities [11]. The crushing, treatment, and preparation of the waste to produce recycled aggregates is currently widespread [12]. In general, aggregates from CDW have a wide field of application in the production of masonry mortars, totally or partially replacing the natural aggregate [13]. Regarding its properties for use in building, it should be highlighted that recycled aggregates have a lower density and a higher coefficient of friability than natural aggregates. This has repercussions in a lower resistance to compression of mortars that incorporate them [14]. The recycled ceramic aggregate has a higher water absorption coefficient compared to recycled concrete aggregate and natural aggregate [15]. This means that the recycled aggregates also have a high water content in fines and other impurities derived from the manufacture [16]. Recycled ceramic aggregate from brick waste and factory works has been proven to have good properties for improving the mechanical resistance of plaster mortars for use in rehabilitation and restoration works of architectural heritage [17]. On the other hand, recycled concrete aggregates have also been used by several authors to increase the density of gypsum compounds, expanding their field of application towards the production of prefabricated building materials [18].

Furthermore, residues from thermal insulation have also been frequently used to improve the physical and mechanical properties of mortars [19]. In the case of expanded polystyrene (EPS), this has been used in various investigations as a substitute for aggregates for the manufacture of mortars with the intention of reducing the final density of the materials and lowering their thermal conductivity [20,21]. It is a material that can be added to the manufacturing process of plaster plates and panels, improving its thermal performance to produce prefabricated elements, although reducing its mechanical resistance to bending [22]. On the contrary, resistance to bending is implemented if insulating mineral wool fibre residues are incorporated [23]. Piña et al. demonstrated the possibility of incorporating this type of crushed waste as a partial replacement for aggregates, presenting good mechanical behaviour and good stability against fire [24,25].

The use of gypsum or plaster mortars to produce prefabricated elements is well known since the incorporation of aggregates improves the mechanical rigidity of the material and increases its resistance [26]. In this sense, Santa Cruz-Astorqui et al. analysed the behaviour of some original $40 \times 20 \times 10$ cm blocks composed of a sandwich panel with two plasterboards and a core of plaster and EPS, showing that this type of prefabricated material has good deformation capacity under external stresses [27]. To improve this deformation capacity, studies have been carried out by various authors in which they have opted for the incorporation of fibres into plaster matrix, together with the incorporation of insulation residues to reduce the final thermal conductivity of the panels produced [28]. Finally, it is worth highlighting the study by Dolezelova et al., where the importance of the shape of aggregates for the manufacture of plaster mortars has been demonstrated, where sands with a rougher particle surface are the ones with a higher quality for the manufacturing of mortars, facilitating the adherence of the conglomerate to its surface and improving resistance to compression [29].

In short, these studies seek to reduce the environmental impact exerted by the construction sector during the process of execution, rehabilitation, and demolition of buildings, seeking ways to prevent and recover CDW in line with the objective established by Directive 2009/98/ EC [30]. The main objective of this paper is to study the technical feasibility of plaster mortars made with recycled aggregate and thermal insulation residues, as no study has been found that shows the possible effect of combining these two types of CDW in plaster mortars for the elaboration of precast concrete products. To do this, an experimental campaign is developed in which, on the one hand, a mechanical and physical characterization is carried out of the material to produce mortars, and, on the other hand, the possible application of these materials for their use in panels and prefabricated building blocks is studied.

## 2. Methodology

This section presents the materials used to manufacture mortars, as well as dosages and a description of the experimental program carried out.

### 2.1. Materials Used

To carry out the experimental campaign of this research, the following raw materials have been used in the preparation of mortars: plaster, water, natural sand, recycled aggregates, and thermal insulation residues (Figure 1).

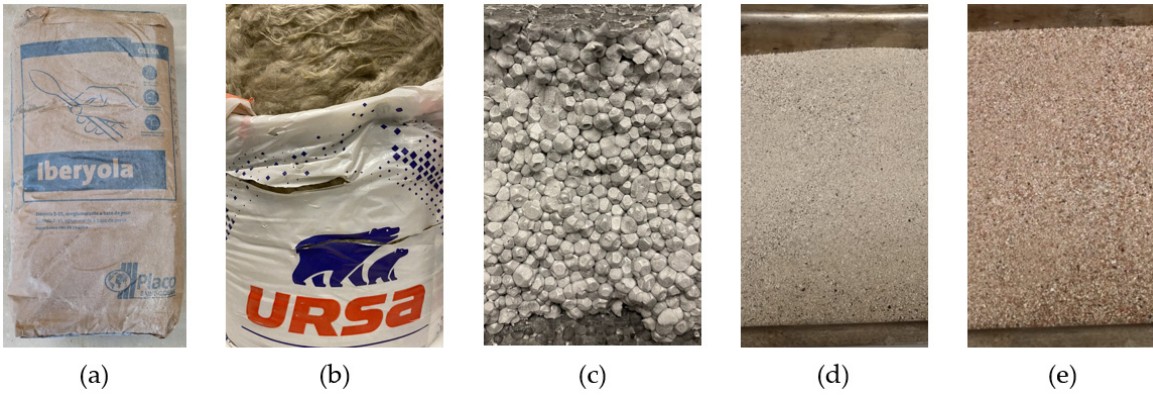

**Figure 1.** (**a**) Plaster E-35; (**b**) mineral wool insulation; (**c**) insulation of expanded polystyrene with graphite; (**d**) recycled concrete aggregate; (**e**) mixed ceramic recycled aggregate.

#### 2.1.1. Binder

E-35 construction plaster was used as a conglomerating material for mixing mortars [31]. It is a material commonly used in construction for wall coverings, the production of prefabricated panels, and the execution of plates for false ceilings [32]. Equations (1) and (2) show the basic scheme of reactions to obtain this raw material [33]:

$$CaSO_4 \cdot 2H_2O \; \rightarrow \; CaSO_4 \cdot \frac{1}{2}H_2O \; (\alpha, \beta) \; + \; \frac{3}{2}H_2O \tag{1}$$

$$CaSO_4 \cdot \frac{1}{2}H_2O \; \rightarrow \; CaSO_4 \; + \; \frac{1}{2}H_2O \tag{2}$$

More specifically, Table 1 shows the characteristics of E-35 plaster used in accordance with the UNE-EN 13279-1:2009 standard [34], which has been supplied by Placo Saint Gobain, S.A. (Madrid Spain). Additionally, X-ray fluorescence analysis revealed that the composition of this raw material is as follows: $CaSO_4$ (99.7%), Al (0.022%), Fe (0.035%), Si (0.068%), Sr (0.157%), and P (0.01%).

**Table 1.** Physical properties of plaster E-35.

| Fire Resistance | Thermal Conductivity | pH | Granulometry | Water Vapor Diffusion Factor | Flexural Strength | Purity Index |
|---|---|---|---|---|---|---|
| Euro class AY | $\lambda = 0.34$ W/mK | >6 | 0–0.2 mm | $\mu = 6$ | >3.5 N/mm$^2$ | >92% |

#### 2.1.2. Thermal Insulation Waste

To improve the thermal behaviour of plaster mortars produced in this research, residues of two types of thermal insulation were used: expanded polystyrene with the addition of graphite and insulating mineral wool. The physical characteristics of these materials provided by URSA Ibérica Aislantes, S.A. (Madrid, Spain), are shown in Table 2.

**Table 2.** Physical properties of the different types of thermal insulation used.

| Insulation | Thermal Conductivity | Density | Water Vapor Diffusion Factor | Geometric Characteristics |
|---|---|---|---|---|
| Expanded Polystyrene with Graphite | $\lambda = 0.031$ W/mK | 28–30 kg/m$^3$ | $\mu = 20$–100 | $\phi = 4$ mm |
| Insulating Mineral Wool | $\lambda = 0.037$ W/mK | 40 kg/m$^3$ | $\mu = 1$ | L = 12 mm |

Before being used in the mortar mix, both types of insulators had to be prepared, as can be seen in Table 2. In the case of expanded polystyrene with graphite, this was separated manually until individual spheres with a mean diameter of four millimetres were obtained, as has been done previously by other researchers [35]. Likewise, mineral wool fibre also had to be crushed and separated manually to a length of 12 mm, following the recommendations of other researchers [36].

2.1.3. Aggregates

Three different types of aggregates have been used in this research: natural aggregate (NA), recycled aggregate from concrete waste (RAcon) and recycled mixed ceramic aggregate from the demolition of masonry walls (RAmix). A physical characterization of these sands for the manufacture of mortars is shown in Table 3.

**Table 3.** Physical characterization of aggregates.

| Test | Fine Content (%) | Particle Form | Fineness Modulus (%) | Friability (%) | Bulk Dens. (kg/m$^3$) | Dry Dens. (kg/m$^3$) | Water Absorption (%) |
|---|---|---|---|---|---|---|---|
| Norma | UNE-EN 933-1 [37] | UNE-EN 13139 [38] | UNE-EN 13139 [38] | UNE-EN 146404 [39] | UNE-EN 1097-3 [40] | UNE-EN 1097-6 [41] | UNE-EN 1097-6 [41] |
| NA | 1.63 | - | 4.47 | 20.43 | 1562 | 2479 | 0.84 |
| RAcon | 3.89 | Not relevant | 4.08 | 23.17 | 1328 | 2246 | 6.43 |
| RAmix | 4.21 | Not relevant | 3.96 | 25.12 | 1297 | 2188 | 7.12 |

A comparison between densities of recycled and natural aggregates can be seen in the analysis in Table 3. Density in recycled aggregates is lower than in natural aggregates. This allows the elaboration of lighter prefabricated elements for use in construction, although it also results in lower mechanical performance [42]. It is also worth noting the high content of fines in these recycled sands, although for the specific case of this investigation, aggregates were sieved and particles retained in the sieves with diameters of 1 mm (60%) and 0.5 mm (40%) were used in order to obtain a homogeneous mixture. Finally, the greater water absorption of these recycled sands is also highlighted, this has its repercussion in a lower workability of mortars during the kneading process, the absorption of the RAmix being greater compared to the RAcon in accordance with other previous studies [43].

Regarding the chemical composition, this was obtained by X-ray fluorescence using Bruker S2 Puma equipment and is shown in Table 4.

**Table 4.** X-Ray fluorescence assay.

| Samples | Al$_2$O$_3$ | CaO | Fe$_2$O$_3$ | K$_2$O | MgO | SiO$_2$ | MnO | TiO$_2$ | SO$_3$ | P$_2$O$_5$ | NaO$_2$ | I.Loss (%) |
|---|---|---|---|---|---|---|---|---|---|---|---|---|
| RAcon | 6.03 | 11.21 | 1.34 | 2.22 | 0.61 | 68.32 | 0.029 | 0.11 | - | 0.12 | 0.35 | 9.66 |
| RAmix | 10.45 | 18.32 | 2.14 | 1.98 | 1.71 | 47.70 | - | 0.34 | 5.37 | 0.12 | 0.63 | 11.24 |

Table 4 shows the higher SiO$_2$ content of RAcon with respect to RAmix, which in turn has a higher CaO and SO$_3$ content because of gypsum impurities derived from the masonry wall cladding [44]. A higher Al$_2$O$_3$ content is also observed in the RAmix because of the ceramic origin of the bricks used in the execution of masonry works [45].

Finally, Table 5 shows the analysis performed by X-ray diffraction with the help of a Siemens D5000 diffractometer with a graphite monochromator Cu-K$\alpha$ ($\lambda$ = 1.540598 Å). The results obtained are shown in Table 5 where it is observed that the predominant crystalline phases for the two types of recycled aggregates used are quartz and calcite [46].

**Table 5.** Analysis by X-ray diffraction, where each (*) shows the relative abundance of each type of mineral crystalline phase.

| Mineral Phase | Calcite | Quartz | Gypsum | Sanidine | Phlogopite |
|---|---|---|---|---|---|
| RAcon | **** | ***** | * | * | * |
| RAmix | **** | ***** | ** | ** | ** |

2.1.4. Water

For the mixing of different dosages, drinking water from Canal de Isabell II (Madrid, Spain) has been used, which has been used successfully in other previous works [47]. The main characteristics of this type of water are its softness (25 mg $CaCO_3$/l) and neutral pH between 7 and 8 [48]. In addition, the following elements can be found in its chemical composition, as listed in Table 6.

**Table 6.** Composition of drinking water in the community of Madrid.

| Nitrate | Nitrite | Free Chlorine | Calcium | Sulphates | Fluorides | Iron | Aluminium |
|---|---|---|---|---|---|---|---|
| 0.60 mg/L | <0.05 mg/L | 0.50 mg/L | 17.80 mg/L | 5.30 mg/L | <0.10 mg/L | 0.01 mg/L | 0.03 mg/L |

*2.2. Experimental Program*

2.2.1. Dosages Used

Throughout this investigation, the following notation presented in Equation (3) has been used to refer to the different types of mortars:

$$E0.8 - Aggregate - Isolation \tag{3}$$

where E0.8 refers to the water/plaster ratio used to prepare the different dosages, Aggregate refers to the type of sand used, which can be of three types: NA (natural sand), RAcon (recycled concrete aggregate) and RAmix (mixed recycled aggregate), and, finally, Isolation refers to the type of thermal insulation waste incorporated: graphite-incorporated expanded polystyrene (EPS) or insulating mineral wool (MW).

For the elaboration of the different types of mortars used in this investigation, the dosages collected in Table 7 have been used. In all cases, the mixtures were carried out following the same techniques and methods that are collected in the UNE-EN standard 12379-2:2014 [49].

**Table 7.** Proportions of each material used in dosages.

| Name | Plaster (g) | Water (g) | Aggregates (g) | EPS (g) | MW (g) |
|---|---|---|---|---|---|
| E0.8 | 1000 | 800 | | | |
| E0.8–NA | 1000 | 800 | 600 | | |
| E0.8–NA–EPS | 1000 | 800 | 600 | 10 | |
| E0.8–NA–MW | 1000 | 800 | 600 | | 7.5 |
| E0.8–RAcon | 1000 | 800 | 600 | | |
| E0.8–RAcon–EPS | 1000 | 800 | 600 | 10 | |
| E0.8–RAcon–MW | 1000 | 800 | 600 | | 7.5 |
| E0.8–RAmix | 1000 | 800 | 600 | | |
| E0.8–RAmix–EPS | 1000 | 800 | 600 | 10 | |
| E0.8–RAmix–MW | 1000 | 800 | 600 | | 7.5 |

It should be noted that in all the mixtures listed in Table 7, residues were manually mixed with plaster powder and then gradually poured into the water to start the mixing process. In addition, samples were kept at a temperature of 22 ± 2 °C and a relative humidity of 60 ± 5%. After seven days of storage under laboratory conditions, samples were dried in an oven at a constant temperature of 40 ± 2 °C for 24 h, as recommended in the UNE-EN 12379-2:2014 standard [49].

### 2.2.2. Instruments and Experimental Plan

In this research, an experimental campaign has been carried out that can be divided into three phases: (1) mechanical characterization, (2) physical characterization of plaster mortars produced, and, later, (3) study of the suitability of these materials for the manufacture of panels and prefabricated blocks. A diagram of the tests carried out is shown in Table 8.

**Table 8.** Planning of the tests carried out in the laboratory.

| Samples | Tests |
|---|---|
| 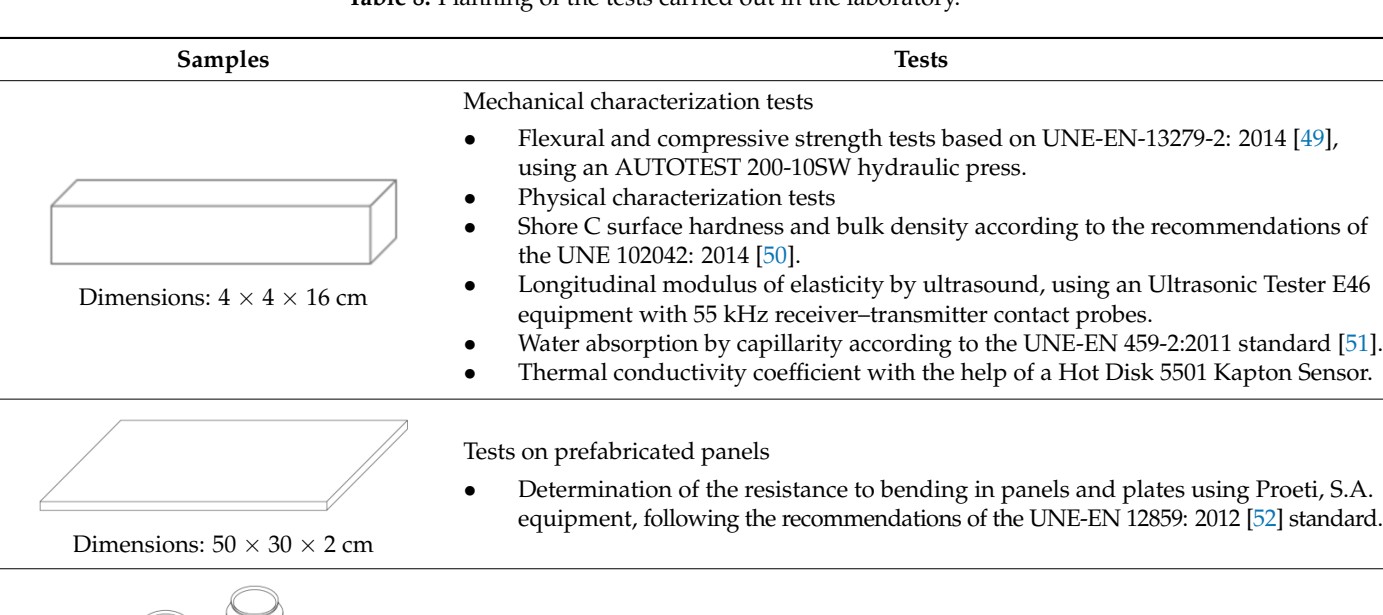Dimensions: 4 × 4 × 16 cm | Mechanical characterization tests<br><br>• Flexural and compressive strength tests based on UNE-EN-13279-2: 2014 [49], using an AUTOTEST 200-10SW hydraulic press.<br>• Physical characterization tests<br>• Shore C surface hardness and bulk density according to the recommendations of the UNE 102042: 2014 [50].<br>• Longitudinal modulus of elasticity by ultrasound, using an Ultrasonic Tester E46 equipment with 55 kHz receiver–transmitter contact probes.<br>• Water absorption by capillarity according to the UNE-EN 459-2:2011 standard [51].<br>• Thermal conductivity coefficient with the help of a Hot Disk 5501 Kapton Sensor. |
| Dimensions: 50 × 30 × 2 cm | Tests on prefabricated panels<br><br>• Determination of the resistance to bending in panels and plates using Proeti, S.A. equipment, following the recommendations of the UNE-EN 12859: 2012 [52] standard. |
| 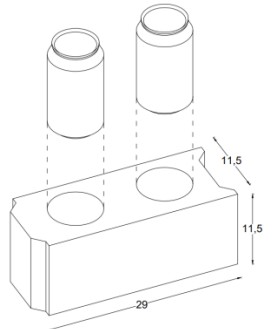Dimensions: 29 × 11.5 × 11.5 cm | Tests on prefabricated blocks<br><br>• Preparation of prefabricated blocks of plaster mortar with recycled aggregate and thermal insulation waste for which some original moulds have been designed, and recycled beverage cans have been used to make the alveoli of the precast and lighten the final weight.<br>• Suction or initial rate of water absorption by capillarity according to the guidelines of the UNE-EN 722-1: 2011 standard [53].<br>• Compressive strength of the blocks using an IBERTEST model MIB-60/AM universal press. |

On the other hand, to determine the effect of the incorporation of sands of a different nature and the different types of thermal insulation residues incorporated in plaster mortars, a study on the analysis of variance (ANOVA) has been carried out. Table 9 shows the factors and levels that have been taken into consideration in the design of experiments.

All the tests performed for statistical analysis have been carried out for a significance level of 5%. For the diagnosis of the model, it has been verified that the residuals of each response variable meet the conditions of normality, homoscedasticity, and independence [54]. A multiple range test has also been included to observe the existence or not of homogeneous groups between the different types of mortars included.

**Table 9.** Factors and levels used for the analysis of variance (ANOVA).

| Factor | Level |
|---|---|
| Aggregate | Natural (NA); Concrete Recycling (RAcon); Mixed Recycling (RAmix) |
| Insulating | Non-Insulating (NI); Expanded Polystyrene (EPS); Mineral Wool (MW) |

## 3. Results and Discussion

This section presents the results derived from the experimental campaign proposed for this study and their discussion.

### 3.1. Mechanical Characterization Tests

Next, Figures 2 and 3 show the results of flexural and compressive strength tests carried out on the 4 × 4 × 16 cm prismatic specimens of plaster mortar.

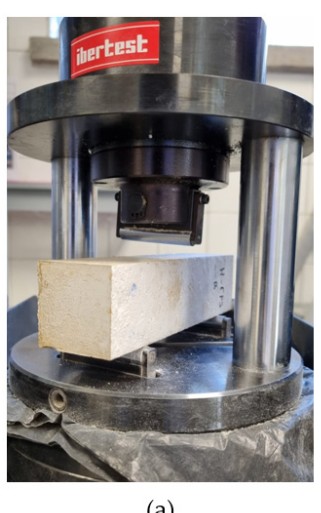

(a)

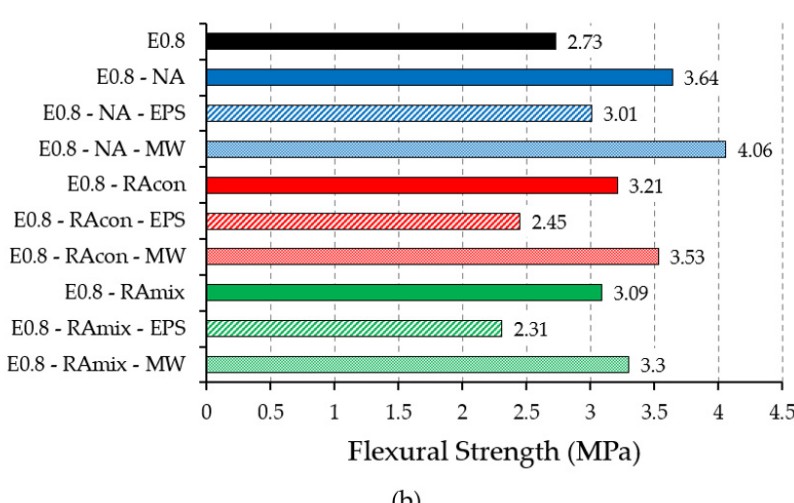

(b)

**Figure 2.** (**a**) Flexural strength test according to UNE-EN-13279-2: 2014; (**b**) results derived from flexural strength test on the prepared mortars.

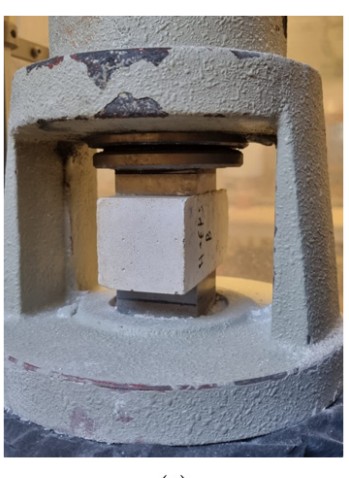

(a)

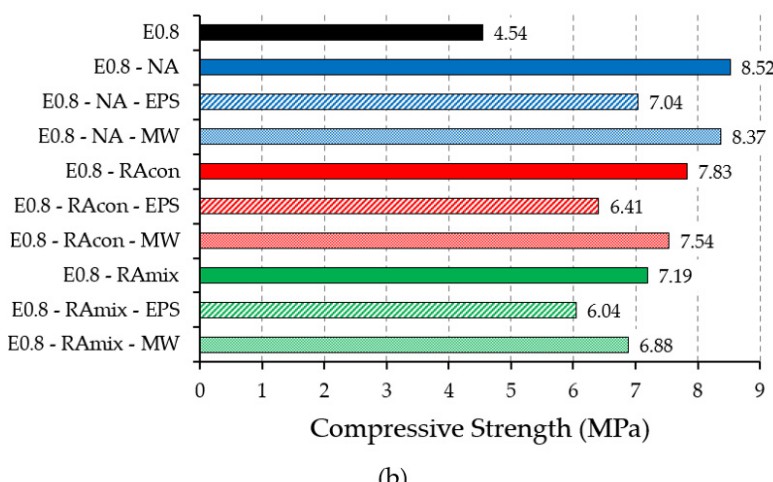

(b)

**Figure 3.** (**a**) Compressive strength test according to UNE-EN-13279-2: 2014; (**b**) results derived from compressive strength test on the prepared mortars.

Figure 2 shows the improvement in flexural strength of plaster materials by incorporating sand into their constitution. All mortars with the incorporation of aggregate without insulation exceeded the E0.8 plaster reference, with mortars made with natural

sand showing the highest values. On the other hand, it can be seen how the incorporation of expanded polystyrene residue with graphite in the manufacture of mortars reduces flexural strength. This is because preferential breakage points occur between EPS spheres and plaster mortar matrix, thus generating greater heterogeneity that negatively affects the mechanical behaviour of the material [55]. On the other hand, the incorporation of mineral wool residue improves the flexural strength of the hardened plaster mortar. This is in line with other studies that highlight the beneficial effect of the incorporation of fibres in the mortar matrix to improve its ductility and deformation capacity [56,57]. Regarding recycled aggregates, mortars that incorporate sand from concrete waste have better performance than mortars with aggregates from mixed ceramic waste.

Furthermore, Figure 3 shows how the incorporation of aggregates improves the compressive strength of plaster material, where mortars made with natural aggregates are the ones that presented the best results. Among mortars with recycled aggregate, the ones that incorporated RAcon obtained the greatest resistance. Finally, it can be seen how the incorporation of EPS residue also decreases the compressive strength of mortars, while the incorporation of mineral wool residue is not decisive in improving this mechanical property [58].

Table 10 shows the results derived from the analysis of variance (ANOVA) carried out to determine the effect of the factors included in the study on the mechanical behaviour of plaster mortars.

**Table 10.** Analysis of variance (ANOVA) for flexural and compressive strength.

| Property | Source | Sum of Squares | Df | Mean Square | F-Ratio | *p*-Value |
|---|---|---|---|---|---|---|
| Flexural Strength (MPa) | **A**: Aggregate | 2.22279 | 2 | 1.11139 | 47.27 | 0.0000 |
| | **B**: Insulating | 5.12703 | 2 | 2.56351 | 109.03 | 0.0000 |
| | **AB**: Interactions | 0.03833 | 4 | 0.00958 | 0.41 | 0.8008 |
| | Residual | 0.42320 | 18 | 0.02351 | | |
| | Total (Corrected) | 7.81134 | 26 | | | |
| Compression Strength (MPa) | **A**: Aggregate | 7.32241 | 2 | 3.66120 | 36.89 | 0.0000 |
| | **B**: Insulating | 9.31770 | 2 | 4.65885 | 46.94 | 0.0000 |
| | **AB**: Interactions | 0.19813 | 4 | 0.04953 | 0.50 | 0.7367 |
| | Residual | 1.78640 | 18 | 0.09924 | | |
| | Total (Corrected) | 18.62460 | 26 | | | |

As can be seen in Table 10, both in mechanical resistance to bending and in compressive strength of mortars, the two factors included in this study (type of aggregate and thermal insulation residue) are statistically significant, having *p*-values lower than the level of significance ($\alpha$ = 0.05).

Finally, Table 11 includes the results obtained for the multiple range test performed for the mechanical properties of mortars.

**Table 11.** Multiple range test for mechanical properties.

| Property | Aggregate | LS Mean | Homogeneous Group | Insulating | LS Mean | Homogeneous Group |
|---|---|---|---|---|---|---|
| Flexural Strength (MPa) | RAmix | 2.900 | X | EPS | 2.590 | X |
| | RAcon | 3.062 | X | NI | 3.314 | X |
| | NA | 3.573 | X | MW | 3.631 | X |
| Comp. Strength (MPa) | RAmix | 6.704 | X | EPS | 6.495 | X |
| | RAcon | 7.260 | X | NI | 7.598 | X |
| | NA | 7.976 | X | MW | 7.847 | X |

In the multiple range test shown in Table 11, it can be seen how there are significant differences for mechanical resistance to bending at all levels for the two factors analysed in

this study. On the other hand, for mechanical compressive resistance, there are significant differences at all levels when we refer to the aggregate type of factor. However, in the case of the incorporation of thermal insulation residues, it cannot be affirmed that there are significant differences between plaster mortars without thermal insulation and those that incorporate mineral wool fibre, both types of mortars presenting greater resistance to statistically significant compression compared to plaster mortars with EPS.

### 3.2. Physical Characterization Tests

This section includes the tests for the physical properties of plaster mortars carried out in this work. These tests have also been carried out on $4 \times 4 \times 16$ cm specimens, and include the following measurements: bulk density, Shore C surface hardness, longitudinal Young's modulus determined by ultrasound, and thermal conductivity coefficient. These are parameters that allow a characterization of the material to later define its possible uses and applications in the building sector.

Figure 4 shows the method used to perform physical characterization tests, and, in Table 12, the results obtained for each of the properties are presented.

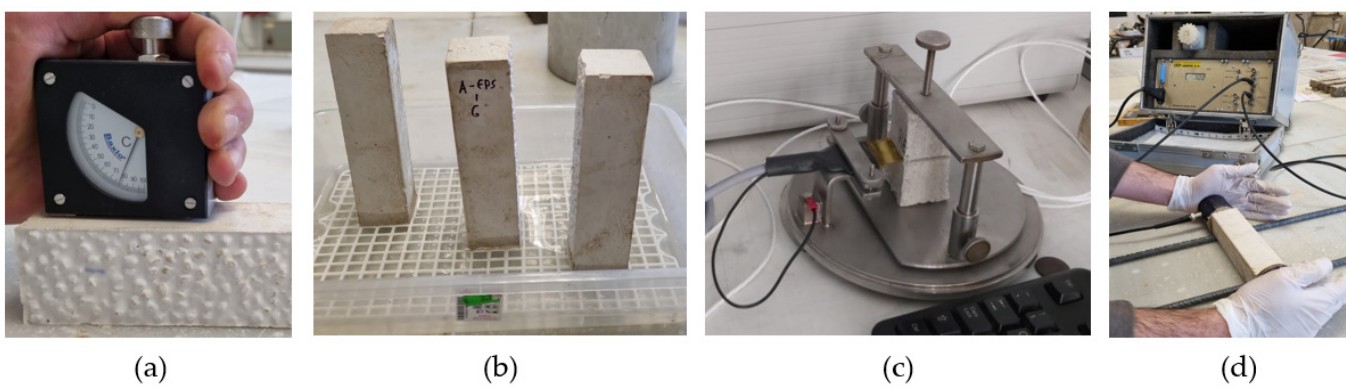

(a)      (b)      (c)      (d)

**Figure 4.** (**a**) Shore C hardness test; (**b**) water absorption by capillarity test; (**c**) determination of the coefficient of thermal conductivity; (**d**) ultrasound test.

**Table 12.** Physical characterization tests of plaster mortars.

| Mortar | Density (kg/m³) | Water Absorption (cm) | Superficial Hardness (Ud. Shore C) | Young Modulus (MPa) | Thermal Conductivity (W/mK) |
|---|---|---|---|---|---|
| E0.8 | 951.7 | 8.8 | 60.7 | 5753.4 | 0.26 |
| E0.8–NA | 1280.0 | 7.4 | 81.3 | 7134.2 | 0.42 |
| E0.8–NA–EPS | 1216.6 | 6.7 | 75.7 | 5599.6 | 0.33 |
| E0.8–NA–MW | 1277.3 | 6.0 | 79.3 | 6044.1 | 0.38 |
| E0.8–RAcon | 1241.7 | 7.8 | 80.7 | 6379.4 | 0.38 |
| E0.8–RAcon–EPS | 1174.0 | 7.1 | 72.7 | 4756.6 | 0.32 |
| E0.8–RAcon–MW | 1233.4 | 7.5 | 77.7 | 5121.3 | 0.36 |
| E0.8–RAmix | 1176.7 | 8.1 | 77.7 | 6117.0 | 0.36 |
| E0.8–RAmix–EPS | 1108.0 | 7.6 | 71.3 | 4491.2 | 0.27 |
| E0.8–RAmix–MW | 1168.7 | 7.7 | 73.7 | 5101.3 | 0.31 |

From the results presented in Table 12, it can be seen how all plaster mortars have a higher density than reference plaster E0.8 due to the incorporation of aggregates in their dosage. In addition, among the elaborated mortars, it can be observed how those that incorporate recycled ceramic aggregate are lighter and how density is reduced when EPS residues are added to the composition of mixtures [59]. On the other hand, absorption of water by capillarity is reduced in plaster mortars. Absorption is lower when mortars are made with natural aggregate compared to those made with recycled aggregate [43]. In addition, in this case, the incorporation of EPS in plaster mortars makes it difficult for the water to rise by capillarity in the materials studied.

In addition, the surface hardness is also increased with the incorporation of sand in plaster mixes, with mortars made with natural aggregate having greater hardness and density being lower in mortars that incorporate EPS. On the other hand, longitudinal Young's modulus determined by ultrasound is also increased in mortars. This is in accordance with the greater mechanical resistance to bending obtained by plaster mortars compared to the reference sample E0.8 [60]. This Young's modulus is greater in mortars made with natural aggregate compared to mortars that incorporate recycled aggregate, furthermore, it decreases when EPS is incorporated in the mortar manufacturing process, while when MW is incorporated, a significant decrease it is noticed. Finally, and in accordance with the results obtained for density, plaster mortars have a higher thermal conductivity than reference E0.8 plaster [61]. However, this thermal conductivity is reduced with the incorporation of thermal insulation residues in mortar mixtures, obtaining lower conductivity values for plaster mortars with EPS than those incorporating MW.

Table 13 shows the results obtained for the analysis of variance (ANOVA) performed for the physical characterization tests, while Table 14 shows the results obtained after performing the multiple range test.

As can be seen in Table 13, all the *p*-values were lower than the level of significance ($\alpha = 0.05$), which implies that both factors included in the design of experiments, type of aggregate and type of insulation, are statistically significant for all the response variables analysed in physical properties of plaster mortars.

**Table 13.** Analysis of variance (ANOVA) for flexural and compressive strength.

| Property | Source | Sum of Squares | Df | Mean Square | F-Ratio | *p*-Value |
|---|---|---|---|---|---|---|
| Density (kg/m$^3$) | **A**: Aggregate | 52,245.9 | 2 | 26,122.9 | 106.72 | 0.0000 |
| | **B**: Insulating | 24,289.4 | 2 | 12,144.7 | 49.62 | 0.0000 |
| | **AB**: Interactions | 37.4815 | 4 | 9.3704 | 0.04 | 0.9969 |
| | Residual | 4406.0 | 18 | 244.778 | | |
| | Total (Corrected) | 80,978.7 | 26 | | | |
| Capillarity water absorption (cm) | **A**: Aggregate | 2.59556 | 2 | 1.29778 | 16.45 | 0.0001 |
| | **B**: Insulating | 1.72222 | 2 | 0.86111 | 10.92 | 0.0008 |
| | **AB**: Interactions | 0.04889 | 4 | 0.01222 | 0.15 | 0.9583 |
| | Residual | 1.42000 | 18 | 0.07889 | | |
| | Total (Corrected) | 5.78667 | 26 | | | |
| Superficial hardness(Shore C) | **A**: Aggregate | 94.889 | 2 | 47.444 | 19.41 | 0.0000 |
| | **B**: Insulating | 200.667 | 2 | 100.333 | 41.05 | 0.0000 |
| | **AB**: Interactions | 8.444 | 4 | 2.111 | 0.86 | 0.5044 |
| | Residual | 44.0 | 18 | 2.444 | | |
| | Total (Corrected) | 348.0 | 26 | | | |
| Young modulus (MPa) | **A**: Aggregate | $5.3566 \times 10^6$ | 2 | $2.6783 \times 10^6$ | 166.64 | 0.0000 |
| | **B**: Insulating | $1.2071 \times 10^7$ | 2 | $6.0356 \times 10^6$ | 375.53 | 0.0000 |
| | **AB**: Interactions | 67,449.1 | 4 | 16,862.3 | 1.05 | 0.4097 |
| | Residual | 289,302 | 18 | 16,072.4 | | |
| | Total (Corrected) | $1.7784 \times 10^7$ | 26 | | | |
| Thermal conductivity (W/mK) | **A**: Aggregate | 0.017785 | 2 | 0.008892 | 18.19 | 0.0000 |
| | **B**: Insulating | 0.028052 | 2 | 0.014026 | 28.69 | 0.0000 |
| | **AB**: Interactions | 0.001948 | 4 | 0.000487 | 1.00 | 0.4350 |
| | Residual | 0.008800 | 18 | 0.000489 | | |
| | Total (Corrected) | 0.056585 | 26 | | | |

**Table 14.** Multiple range test for physical properties.

| Property | Aggregate | LS Mean | Homogeneous Group | | Insulating | LS Mean | Homogeneous Group | |
|---|---|---|---|---|---|---|---|---|
| Density (kg/m³) | RAmix | 1151.11 | X | | EPS | 1166.22 | X | |
| | RAcon | 1216.22 | | X | MW | 1226.44 | | X |
| | NA | 1258.00 | | X | NI | 1232.78 | | X |
| Water absorption (cm) | NA | 7.022 | X | | EPS | 7.144 | X | |
| | RAcon | 7.367 | X | | MW | 7.367 | X | |
| | RAmix | 7.778 | | X | NI | 7.756 | | X |
| Hardness Shore C | RAmix | 74.2 | X | | EPS | 73.2 | X | |
| | RAcon | 77.0 | | X | MW | 76.9 | X | |
| | NA | 78.8 | | X | NI | 79.9 | | X |
| Young modulus (MPa) | RAmix | 5236.48 | X | | EPS | 4948.97 | X | |
| | RAcon | 5418.96 | | X | MW | 5422.24 | X | |
| | NA | 6259.28 | | X | NI | 6543.50 | | X |
| Thermal Conduct. (W/mK) | RAmix | 0.3156 | X | | EPS | 0.3089 | X | |
| | RAcon | 0.3544 | | X | MW | 0.3511 | X | |
| | NA | 0.3778 | | X | NI | 0.3878 | | X |

Table 14 shows the composition of homogeneous groups after the multiple range test. This table shows how mortars with natural aggregate have a higher surface hardness and a higher Young's modulus, in accordance with Table 12, and how the incorporation of thermal insulation residues decreases the values obtained in these physical properties. In regard to thermal conductivity, mortars with recycled aggregate have better performance for this property, reducing this conductivity with the incorporation of thermal insulation waste and especially EPS [62]. Finally, with respect to apparent density, it can be seen how mortars with recycled aggregate are lighter. This affects the results of the compressive strength test, but it can also be seen that there are no differences between homogeneous groups in mortars with mineral wool insulation and without insulation. Likewise, in the absorption of water by capillarity, mortars with a natural aggregate present the best behaviour. It can also be observed how the incorporation of thermal insulation residues reduces the height reached by the water in this test with respect to mortars without insulation.

*3.3. Tests on Prefabricated Plates and Blocks*

3.3.1. Prefabricated Blocks

A possible use of these plaster mortars is the production of prefabricated blocks for construction. This section presents the results obtained after the tests carried out on blocks of 29 × 11.5 × 11.5 cm, where the alveoli to lighten weight have been made by incorporating two recycled soft drink cans with a diameter of 6.5 cm and a height of 11.5 cm. The template and test blocks used are shown in Figure 5.

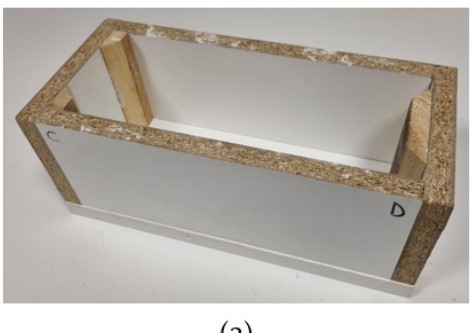
(a)

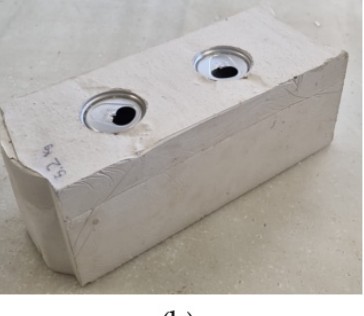
(b)

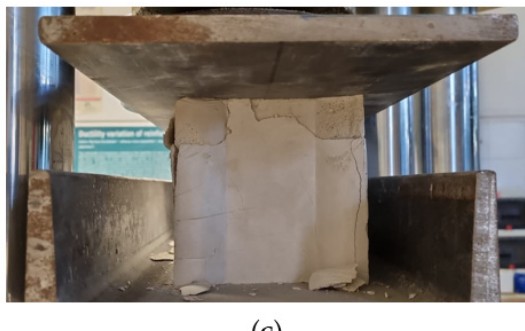
(c)

**Figure 5.** (**a**) Mould for the elaboration of prefabricated blocks; (**b**) type prefabricated block; (**c**) test of resistance to compression of the blocks.

The capillarity water absorption tests are shown in Table 15 for the different blocks produced.

**Table 15.** Initial rate of water absorption by capillarity according to the guidelines of the UNE-EN 722-1: 2011 standard.

| Aggregates | NA | | | RAcon | | | RAmix | | | E0.8 |
|---|---|---|---|---|---|---|---|---|---|---|
| Insulation | NI | EPS | MW | NI | EPS | MW | NI | EPS | MW | |
| Absorption (kg/mm² min) | 4.89 | 4.53 | 4.68 | 5.17 | 4.95 | 5.12 | 5.36 | 5.14 | 5.31 | 7.03 |

As can be seen in Table 15, the incorporation of sand in plaster material reduces the water absorption coefficient compared to the reference material E.08. Mortars with the incorporation of RAmix presented the highest absorption coefficient [63], and, in all cases, the incorporation of thermal insulation residues has reduced this absorption coefficient.

Figure 6 shows the results obtained after the compressive strength test on mortar blocks.

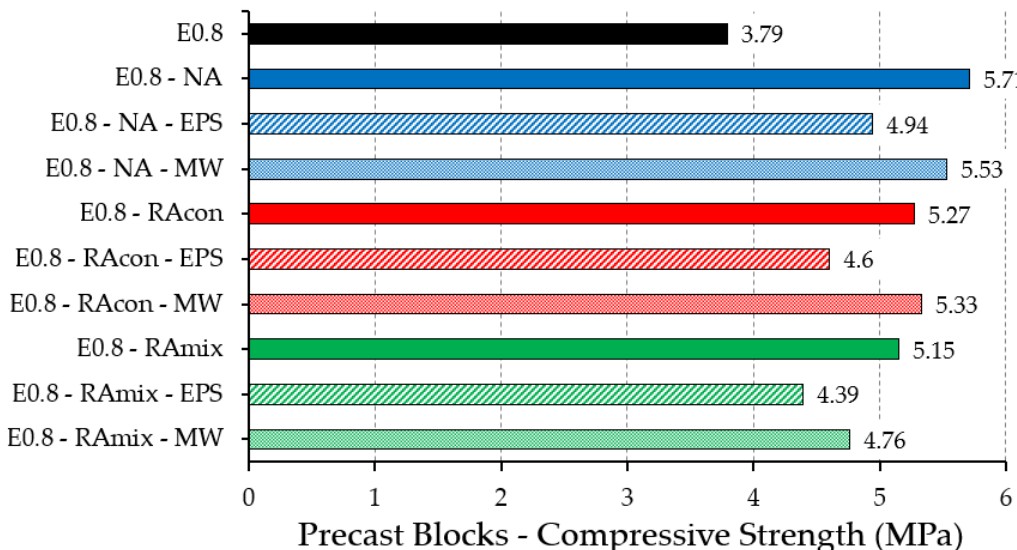

**Figure 6.** Results of compressive strength test on prefabricated blocks.

As can be seen in Figure 6, all plaster mortar blocks exceeded in their mechanical capacity the compressive strength of the reference block E0.8. The results are in accordance with those presented in Figure 3, where mortars made with natural aggregate have higher resistance than mortars made with RAcon, and these in turn have higher resistance than those made with RAmix. It is also observed how the incorporation of thermal insulation residues reduces compressive strength, especially in the case of expanded polystyrene with graphite. Finally, it should be noted that in all the cases tested, plaster mortar presented good adherence to the cans used to make the alveoli, even so, in all the blocks the effect of the breakage in the core of the precast was observed, coinciding with these alveoli that are the most fragile areas and therefore the preferred points of breakage.

### 3.3.2. Prefabricated Plates

This section includes the results derived from the tests on $50 \times 30 \times 2$ cm mortar plates. Figure 7 shows the test method used.

These tests on plates and panels are of special importance to study the behaviour of plaster mortars made in prefabricated pieces with dimensions close to the real ones used in the study, as well as to evaluate their deformation capacity under possible alterations of the structure [64]. The results obtained for the different types of plates made in this study are shown in Figure 8.

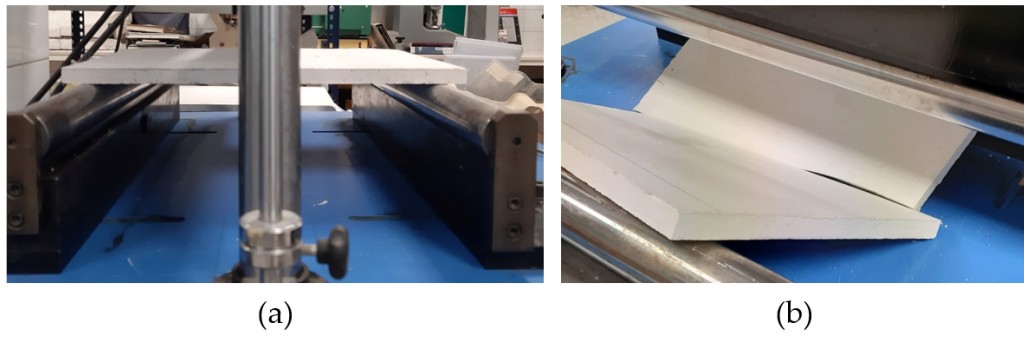

(a)            (b)

**Figure 7.** Flexural strength test on prefabricated panels. (**a**) Plate before the test; (**b**) plate after assay.

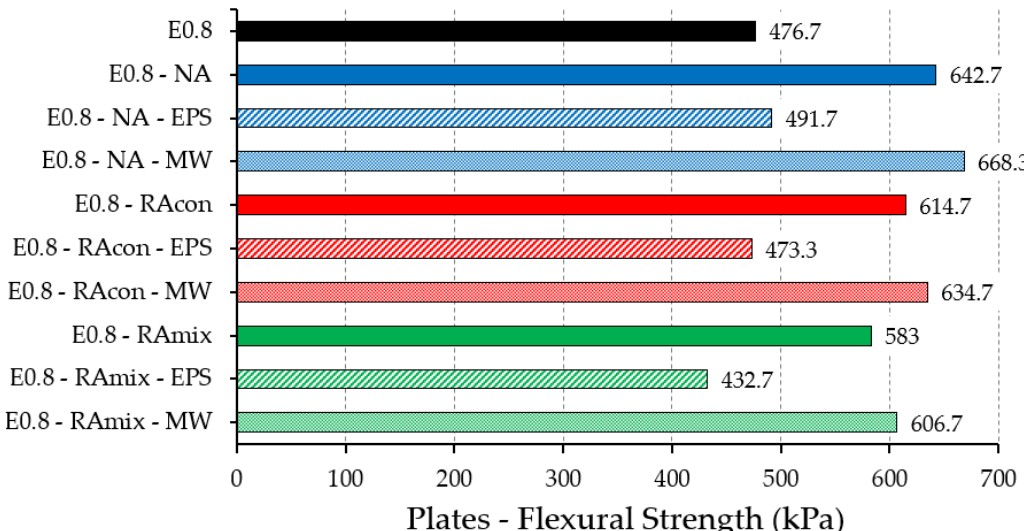

**Figure 8.** Results of flexural strength test on prefabricated plates.

Figure 8 shows how flexural strength is reduced in plates and panels with dimensions close to reality, in comparison with standardized RILEM $4 \times 4 \times 16$ cm specimens studied in Section 3.1. Likewise, it is observed how flexural strength is increased by incorporating aggregates in the matrix of plaster material. In addition, and as expected, mortars that incorporate mineral wool fibres have a greater resistance to bending and are more suitable for the manufacture of prefabricated panels and plates. On the other hand, mortars with the incorporation of expanded polystyrene residue with graphite showed lower flexural strengths, and mortars made with RAmix–EPS show strengths below the reference E0.8 plaster.

The results obtained in this research should be understood as a characterization of materials tested. If these materials were to be used in real on-site prefabricated elements, mechanical tests should be carried out on pieces with the specific geometry of the slabs and panels in order to consider the size effect. In addition, it is recommended to use these materials in interiors and not as cladding mortars on external facades. However, it would be advisable in future studies to carry out research that considers the durability of these materials, such as: humidity and drying cycles, total water absorption, thermal shock, etc. Thus, a future line of research to improve the technical performance of these materials involves the incorporation of reinforcement fibres in plaster matrix.

## 4. Conclusions

From the results obtained in this work, it can be concluded that recycled aggregates and thermal insulation residues for the elaboration of plaster mortars can be applied in building and are more respectful to the environment. In particular, the following conclusions were reached:



- With regard to the mechanical properties, these are implemented in plaster mortars with respect to the reference material E0.8. Mortars with natural aggregate offer better mechanical behaviour than mortars with recycled aggregate, where mortars with RAmix obtained the lowest compressive and flexural strength. In the specific case of flexural strength, it has been possible to observe the positive effect offered by the incorporation of mineral wool insulation fibre, increasing the flexural strength of mortars. In turn, compressive strength is reduced when EPS residues are incorporated into the plaster mortar matrix.
- The density of plaster mortars is much higher than that obtained by the reference plaster E0.8, although the incorporation of EPS waste manages to lighten the weight of these construction materials.
- The incorporation of EPS residues in the manufacture of mortars reduces the height reached by the water in the capillary absorption test in plaster mortars. Mortars that presented the greatest absorption were those made with RAmix. Any of them exceeded the reference E0.8 plaster values.
- In turn, mortars with the incorporation of EPS residue presented the lowest coefficient of thermal conductivity. Additionally, in all cases, it was possible to observe the beneficial effects of incorporating thermal insulation residues to improve this property of mortars.
- In terms of surface hardness and Young's modulus, plaster mortars made with natural aggregate obtained the best results. In addition, in all cases, the incorporation of thermal insulation residues reduces the values obtained for these physical properties.
- Regarding the viability of plaster mortars to make plates and panels, it has been possible to verify how the incorporation of mineral wool fibre in its composition improves the final resistance to bending of the precast.
- Finally, some prefabricated blocks of plaster mortar of our own design have been made. These presented a good resistance to compression and the best results were obtained for mortars with natural sand and without thermal insulation residues.

In general, it can be seen how plaster mortars have good technical performance for their application in building, and how the incorporation of recycled aggregates and thermal insulation residues improves the mechanical and physical properties of traditional plasters. The performance of this type of study is in line with the Sustainable Development Goals set by the United Nations Organization, supporting a more efficient use of natural resources and a decrease in the consumption of raw materials.

**Author Contributions:** Conceptualization, D.F. and M.Á.; methodology, D.F.; software, D.F. and P.S.; validation, P.S., D.F. and A.Z.; formal analysis, D.F. and P.S.; investigation, D.F. and M.Á.; resources, D.F.; data curation, D.F. and P.S.; writing—original draft preparation, D.F. and A.Z.; writing—review and editing, M.Á.; visualization, A.Z.; supervision, D.F.; project administration, D.F.; funding acquisition, D.F. All authors have read and agreed to the published version of the manuscript.

**Funding:** This research was funded by URSA Ibérica Aislantes, S.A., grant number P2054090068.

**Institutional Review Board Statement:** Not applicable.

**Informed Consent Statement:** Not spplicable.

**Data Availability Statement:** Not applicable.

**Acknowledgments:** The authors would like to acknowledge the collaboration of the company URSA Ibérica Aislantes, SA, through the project P2054090068 "Thermo-acoustic solutions in housing renovation, simulation, and monitoring", which has served as a support and initiative framework for the realization of this research.

**Conflicts of Interest:** The authors declare no conflict of interest.

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
