# Peer review of "Experimental Study with Plaster Mortars Made with Recycled Aggregate and Thermal Insulation Residues for Application in Building"

_sustainability, doi:10.3390/su14042386_

Round 1

Reviewer 1 Report

The paper is about an interesting topic which is the use of raw materials from construction and demolition waste for a sustainable building sector. Authors have incorporated recycled aggregate and thermal insulation residues to a plaster mortar. Then, they have studied the mechanical and physical properties of the mixes. They have also studied the suitability of using of the materials for panels and prefabricated building blocks. The research is succinctly described and contextualized. Argumentation is consistent with the results obtained and those referenced in the literature which is up to date. 

The authors should also argument some of their findings. For example, in line 141-143: The aggregate used were of 1mm diameter max; how can the authors explain this choice? Can this have an impact on the density of the material used?

Sometimes the values of results are very closed to each other, can the authors provide standards deviation for them? In order to better observe the influence of incorporating the different materials

Author Response

The paper is about an interesting topic which is the use of raw materials from construction and demolition waste for a sustainable building sector. Authors have incorporated recycled aggregate and thermal insulation residues to a plaster mortar. Then, they have studied the mechanical and physical properties of the mixes. They have also studied the suitability of using of the materials for panels and prefabricated building blocks. The research is succinctly described and contextualized. Argumentation is consistent with the results obtained and those referenced in the literature which is up to date. 

The authors are grateful for the reviewer's comments and fully agree with his observations. The proposed minor changes have been made.

The authors should also argument some of their findings. For example, in line 141-143: The aggregate used were of 1mm diameter max; how can the authors explain this choice? Can this have an impact on the density of the material used?

The aim was to recycle the fine fraction of the aggregates and observe its impact on the density in a mix that was as homogeneous as possible. Coarser aggregate fractions would have provided higher mechanical compressive strengths, but would have limited the thickness of the precast slabs.

Sometimes the values of results are very closed to each other, can the authors provide standards deviation for them? In order to better observe the influence of incorporating the different materials

The standard deviation has not been included in order not to be redundant, as it is used for the analysis of homogeneous groups and is implicit in the tables for the multiple range test.

Reviewer 2 Report

Investigating the sustainable usage of CDW is a buzzing field of research. Authors made a significant attempt to investigate the plaster mortars properties prepared with CDW. Very well presented and nicely documented. I have proposed a few minor/moderate editorial and technical corrections/explanations. If the following suggestions are incorporated, I believe this manuscript has the potential to be published at 'Sustainability'.

[1] Usage of article needed to be revised, mostly the usage of 'the' followed by plurals. Please correct all the punctuation error. For instance, double full stop (..), non-capital letter after full stop, missing commas and semicolons, etc.  
[2] If Table 4 is entirely obtained from the literature, please put the reference at the Table 4 title. For instance, Table 4. X-Ray Fluorescence Assay [42, 43]. 
[3] Line 154 - It says 'Table X'. I believe it should be Table 5.
[4] It's not clear from the Table 5, what **** and * means? Mention it in the text to increase the readability.
[5] Title of the Table 7 is too short and vague. Elaborate it. 
[6] Line 190 - "A diagram of he tests carried out ....", fix the typo in this sentence. Should be 'the' instead of 'he'!
[7] Table 10 - what is the significance of F-Ratio? To check if the data are statistically significant, p-value might be sufficient? Ditto for Table 13. 
[8] Table 11 - what are the meaning of X? Can you please explain a bit about the homogeneous group in the text? It's not clear what does it mean when they are in the same line or diagonal? Ditto for Table 14.
[9] After you explain the Figure 8, this paper suddenly finished! Please add a section/subsection by explaining how the outcome of this research can address the research gap you mentioned in the introduction? how the outcomes can be used in the construction industry? what are your recommendations, suggestions based on these findings? what are the future options to explore? etc.

Author Response

Investigating the sustainable usage of CDW is a buzzing field of research. Authors made a significant attempt to investigate the plaster mortars properties prepared with CDW. Very well presented and nicely documented. I have proposed a few minor/moderate editorial and technical corrections/explanations. If the following suggestions are incorporated, I believe this manuscript has the potential to be published at 'Sustainability'.

The authors are grateful for the reviewer's comments and fully agree with his observations. The proposed minor changes have been made.

[1] Usage of article needed to be revised, mostly the usage of 'the' followed by plurals. Please correct all the punctuation error. For instance, double full stop (..), non-capital letter after full stop, missing commas and semicolons, etc.

Language and punctuation of all the paper have been reviewed.

 [2] If Table 4 is entirely obtained from the literature, please put the reference at the Table 4 title. For instance, Table 4. X-Ray Fluorescence Assay [42, 43].

This test was carried out by the authors and are concrete results of the aggregates used in the research.

[3] Line 154 - It says 'Table X'. I believe it should be Table 5.

Correct, the error has been corrected

[4] It's not clear from the Table 5, what **** and * means? Mention it in the text to increase the readability.

It relates to the relative abundance of each raw material, this is a qualitative study that is used in some work with recycled aggregates. It has been mentioned in the text according to the review.

[5] Title of the Table 7 is too short and vague. Elaborate it. 

The title has been improved.

[6] Line 190 - "A diagram of the tests carried out ....", fix the typo in this sentence. Should be 'the' instead of 'he'!

The error in the text has indeed been corrected.

[7] Table 10 - what is the significance of F-Ratio? To check if the data are statistically significant, p-value might be sufficient? Ditto for Table 13. 

Indeed, the p-value would be sufficient. The authors only wanted to provide the value of the contrast statistic used in the analysis.

[8] Table 11 - what are the meaning of X? Can you please explain a bit about the homogeneous group in the text? It's not clear what does it mean when they are in the same line or diagonal? Ditto for Table 14.

The homogeneous group analysis analyses the differences between the different mixes included in the study. The interpretation of this is as follows: when the X's are aligned one below the other on the same vertical, it means that there is no significant difference between those groups of mortars. However, when they are diagonally and not aligned, it means that the difference is statistically significant between those groups. This information has been clarified in the text.

[9] After you explain the Figure 8, this paper suddenly finished! Please add a section/subsection by explaining how the outcome of this research can address the research gap you mentioned in the introduction? how the outcomes can be used in the construction industry? what are your recommendations, suggestions based on these findings? what are the future options to explore? etc.

The paragraph mentioned above has been added.

Reviewer 3 Report

Manuscript ID: sustainability-1600817

Title: Experimental Study with Plaster Mortars Made with Recycled Aggregate and Thermal Insulation Residues for Application in Building

The study is quite self-sufficient, complete and relevant to sustainability. However, some revision is necessary for this paper to get accepted for publication.

Introduction section needs to show a broader perspective. For example, more examples of sustainable cements like wollastonite mineral based cements [a] and flyash based geopolymer cements [] should be discussed with appropriate references. [a] Dey, T. Thermal Properties of a Sustainable Cement Material: Effect of Cure Conditions. Ceramics-Silikáty, 2014, 58(4), 275-281.

Also more examples of sustainable and recycled thermal insulation should be discussed with proper references, such as aerogel [], foam glass [] etc.

Section 2.2: The first subsection here should be titled as ‘Instruments’ where the brand and model of all instruments should be mentioned along with basic operating conditions.

Line 154: What is Table X? Will X be replaced by some number?

Table 12: What is Tipo?

Author Response

The study is quite self-sufficient, complete and relevant to sustainability. However, some revision is necessary for this paper to get accepted for publication.

The authors are grateful for the reviewer's comments and fully agree with his observations. The proposed minor changes have been made.

Introduction section needs to show a broader perspective. For example, more examples of sustainable cements like wollastonite mineral based cements [a] and fly ash based geopolymer cements [] should be discussed with appropriate references. [a] Dey, T. Thermal Properties of a Sustainable Cement Material: Effect of Cure Conditions. Ceramics-Silikáty, 2014, 58(4), 275-281.

Also more examples of sustainable and recycled thermal insulation should be discussed with proper references, such as aerogel [], foam glass [] etc.

The authors understand that the article is focused on plaster and specific types of thermal insulation that have been included in the introduction by carrying out a bibliographic search in specialised databases. We understand that diverting attention to other materials can be counterproductive for the follow-up of the article and tedious in the writing, however, the ideas raised by the reviewer will be taken into account for future research in the area of cement mortars or mortars with other types of different thermal insulation residues.

Section 2.2: The first subsection here should be titled as ‘Instruments’ where the brand and model of all instruments should be mentioned along with basic operating conditions.

The name of the section has been changed and the brand and model of all instruments of the known elements have been added.

 Line 154: What is Table X? Will X be replaced by some number?

Correct, the error has been corrected

Table 12: What is Tipo?

Translation error corrected

Reviewer 4 Report

I had following clarification regarding article

  1. Abstract need to improve. abstract should include a small introduction about topic, need for research, objective of research, experiment and output of research in 150 words.
  2. Introduction: include 6 factors that influence on carbon foot print
  3. Some sentences are too long and using difficult words to understand. Authors are requested to use simple English for example page number 2: line 4-9.
  4. Page 2: paragraph 2: authors are requested to add more references for "addition of recycled materials reduce final density of composite thereby reducing thermal conductivity"
  5. Page 2: paragraph 3: line 4-6: rewrite the sentence. What kind of absorption?
  6. Authors are requested to add research gap in introduction in a clear way
  7. Authors are requested to add the 'Reasons for low thermal conductivity of composite material with recycled materials' in introduction part
  8. Objective of article need to be rewritten
  9. Thermal insulation residue is a waste? How it should be extracted from a waste? Add references to it.
  10. Give an introduction/application about E35 plaster with reference
  11. Rewrite the first line of first paragraph of 4th page
  12. Add particle size distribution for the ingredients used in this study
  13. Rewrite the first line of third paragraph of 4th page
  14. Check the nomenclature of the article throughout
  15. Add XRD curves for recycled materials
  16. Page 5, add what is the type of ultrasonic testing used in this research? Direct / semi-direct/
  17. Page 6: add references for comparison with mortar with natural aggregates and recycled aggregates

Author Response

I had following clarification regarding article

Abstract need to improve. abstract should include a small introduction about topic, need for research, objective of research, experiment and output of research in 150 words.

Abstract has been corrected in line with the reviewer's comments.

Introduction: include 6 factors that influence on carbon foot print

The above-mentioned corrections have been included

Some sentences are too long and using difficult words to understand. Authors are requested to use simple English for example page number 2: line 4-9.

Sentence have been rewritten

Page 2: paragraph 2: authors are requested to add more references for "addition of recycled materials reduce final density of composite thereby reducing thermal conductivity"

A further reference has been added in line with the reviewer's comment.

Page 2: paragraph 3: line 4-6: rewrite the sentence. What kind of absorption?

Sentence have been corrected

Authors are requested to add research gap in introduction in a clear way

The research gap in the introduction has been better highlighted.

Authors are requested to add the 'Reasons for low thermal conductivity of composite material with recycled materials' in introduction part

The answer to this answer is resolved by question 4 when it is stated that this is due to its lower density, as pointed out by the reviewer.

Objective of article need to be rewritten

Paragraph have been changed

Thermal insulation residue is a waste? How it should be extracted from a waste? Add references to it.

This is the waste generated during the application of the thermal insulation of the façades. The preparation of these wastes for use in the mortars used in the study was carried out manually as described in the methodology.

Give an introduction/application about E35 plaster with reference

E-35 plaster reference have been included in line with reviewer’s comment.

Rewrite the first line of first paragraph of 4th page

Sentence have been rewritten

Add particle size distribution for the ingredients used in this study

Information has been added in the text

Rewrite the first line of third paragraph of 4th page

Sentence have been rewritten

Check the nomenclature of the article throughout

Nomenclature has been revised in line with the reviewer's comment.

Add XRD curves for recycled materials

These curves are not available. Only qualitative analysis has been carried out for the relative abundance of each material.

Page 5, add what is the type of ultrasonic testing used in this research? Direct / semi-direct/

A direct test has been carried out. This information has been added to the document.

Page 6: add references for comparison with mortar with natural aggregates and recycled aggregates

No studies have been found on plaster mortars that are similar to the one presented in this research and can be referenced in this section.